# Fish CDK2 recruits Dtx4 to degrade TBK1 through ubiquitination in the antiviral response

Long-Feng Lu[1,2], Can Zhang[1,2], Zhuo-Cong Li[1,2], Bao-Jie Cui[1,3], Yang-Yang Wang[1,2], Ke-Jia Han[1,3], Xiao Xu[1,3], Chu-Jing Zhou[1,3], Xiao-Yu Zhou[1,2], Yue Wu[1,2], Na Xu[1,2], Xiao-Li Yang[1,2], Dan-Dan Chen[1,2], Xiyin Li[1,2,4,5], Li Zhou[1,2,4,5], Shun Li[1,2,4,5,6]*

[1]Institute of Hydrobiology, Chinese Academy of Sciences, Wuhan, China; [2]University of Chinese Academy of Sciences, Beijing, China; [3]College of Fisheries and Life Science, Dalian Ocean University, Dalian, China; [4]State Key Laboratory of Freshwater Ecology and Biotechnology, Institute of Hydrobiology, Wuhan, China; [5]Key Laboratory of Aquaculture Disease Control, Ministry of Agriculture, Wuhan, China; [6]Laboratory for Marine Biology and Biotechnology, Qingdao Marine Science and Technology Center, Qingdao, China

*For correspondence: bob@ihb.ac.cn

## eLife Assessment

This **valuable** study uses zebrafish as a model to reveal a role for the cell cycle protein kinase CDK2 as a negative regulator of type I interferon signaling. The evidence supporting the authors' claims is **convincing**, including both in vivo and in vitro investigative approaches that corroborate a role for CDK2 in regulating TBK1 degradation. In this latest version, the authors included data addressing a concern raised by the reviewer in the previous peer review round. This work will interest cell biologists, immunologists, and virologists.

**Abstract** Although the classical biological protein cell cycle protein kinase CDK2 has been extensively studied in higher vertebrates, its function in lower vertebrates beyond the regulation of mitosis remains unknown. In this study, we report a distinct mechanism whereby IFN expression is negatively regulated in fish by CDK2. After infection with the spring viremia of carp virus (SVCV), fish CDK2 expression significantly increased in tissues and cells. Moreover, antiviral resistance was improved in *cdk2*[-/-] homozygotes, and the antiviral cytokine interferon (IFN) expression was significantly higher. At the cellular level, CDK2 overexpression reduced IFN expression, while *cdk2* knockdown increased the ability of cells to produce IFN. Subsequently, it was discovered that fish CDK2 binds and degrades TBK1, resulting in reduced IFN. CDK2 increases the K48-linked ubiquitination of TBK1, causing its degradation, while E3 ubiquitin ligase Dtx4 was found to be involved in this process following the significant enhancement of TBK1 K48-linked ubiquitination. Protein mass spectrometry and immunoblot analysis confirmed that the K567 site on TBK1 is essential for CDK2 to engage with Dtx4 and degrade TBK1; thus, after mutating the K567 site, K48-linked ubiquitination of TBK1 was not enhanced by Dtx4, and TBK1 was not degraded by CDK2. Our data demonstrate that fish CDK2 recruits the E3 ubiquitin ligase Dtx4 to target the K567 site of TBK1 and promote its degradation. These results suggest that CDK2 in lower vertebrates is implicated in a specialized role for antiviral innate immunity.

## Introduction

Interferon (IFN) production is central to the host's innate immune response to viral infections. IFN acts as a ligand, binding to receptors on neighboring cell membranes through the autocrine and paracrine pathways, which initiates the Janus kinase/signal transducer and activator of the transcription (JAK/STAT) signaling pathway (*Kisseleva et al., 2002*). This, in turn, leads to the transcription of a large number of antiviral genes, ultimately resulting in the clearance of intracellular viral components (*Sadler and Williams, 2008*). IFN induction depends on cellular pattern recognition receptors (PRRs) and sensing of conserved pathogen-associated molecular patterns (PAMPs), with the retinoic acid-inducible gene I (RIG-I)-like receptors (RLRs) signaling pathway being critical in this system (*Kumar et al., 2011*). Viral nucleic acids are recognized by the RLRs, which then initiate a series of signaling events that lead to downstream IFN expression. TANK-binding kinase 1 (TBK1) efficiently phosphorylates IFN regulatory factor 3/7 (IRF3/7), facilitating its nuclear entry (*Fitzgerald et al., 2003*).

Effective control of IFN production is crucial during viral infection in maintaining immune responses and homeostasis, whereby excessive IFN can cause inflammatory storms that harm the organism. Hence, TBK1 plays a significant role in the IFN signaling pathway and is negatively regulated by various factors to balance IFN expression in the host (*Zhao, 2013*). Mammalian TBK1 is degraded by NLRP4, which recruits the E3 ubiquitin ligase DTX4 for K48-linked polyubiquitination at Lys670 (*Cui et al., 2012*). Additionally, DYRK2 is crucial in the degradation of TBK1 by NLRP4 through the phosphorylation of Ser527, as mentioned in previous studies (*Cui et al., 2012*; *An et al., 2015*). The function of TBK1 is conserved in fish and subject to modulation by multiple molecules to prevent excessive IFN expression. Indeed, TMEM33 in zebrafish serves as a TBK1 substrate, reducing IRF3 phosphorylation and hindering TBK1 kinase activity to diminish IFN expression (*Lu et al., 2021*). Generally, the ability of TBK1 to induce IFN is conserved and tightly regulated within the host.

CDKs are a family of Ser/Thr kinases that act as cell cycle regulators. Cell cycle progression from the G1 phase to the S phase and from G2 to mitosis is controlled by various cyclin–CDK complexes. For example, CDK2 and cyclin E regulate the G1–S transition (*Fagundes and Teixeira, 2021*). Further studies have unveiled other functions of the CDK family in biological processes beyond cell cycle regulation. The transcription of multiple proinflammatory genes is upregulated in a CDK-dependent fashion throughout the G1 phase (*Schmitz and Kracht, 2016*). In the context of antitumor immunity in fibrosarcoma and lung carcinoma, inhibiting CDK2 leads to the RB protein being phosphorylated, which results in increased production of type I IFN (*Chen et al., 2022*). Another study demonstrated that inhibiting CDKs or the knockdown of CDKs activities caused a significant blockade in IFN release from culture supernatants (*Cingöz and Goff, 2018*). While there are several reported functions of CDKs in innate immunity, understanding the roles of CDKs in this area remains unclear.

The immune systems of higher vertebrates (e.g. humans) and lower vertebrates (e.g. fish) generally exhibit some consistency, although there are notable differences. For instance, IFN phosphorylates fish IRF3, which is only phosphorylated by viruses in mammals. Additionally, fish MVP functions as an IFN-negative regulator, while human MVP mediates IFN-positive expression (*Sun et al., 2010*; *Panne et al., 2007*; *Li et al., 2019*; *Liu et al., 2012*). There are 13 CDK genes in the human genome and 21 CDK orthologs in zebrafish, indicating that CDK functions are conserved and differ between fish and humans. Our report reveals that fish CDK2 acts as an IFN negative regulator that recruits Dtx4 to facilitate the ubiquitination of TBK1, thereby restricting IFN expression. These findings offer valuable insights into the various roles of CDK2.

## Results

### CDK2 is upregulated during viral infection in vivo and in vitro

To identify potential molecules linked with viral infection, we generated a transcriptome pool from liver and spleen tissues taken from SVCV-infected zebrafish. Notably, classical cell cycle regulator *cdk2* was among the few upregulated genes in the CDK family (*Figure 1A and B*). In previous studies, it was found that mammalian *cdk2* was not regulated during viral infection. As shown in *Figure 1C*, this was confirmed by detecting *cdk2* after THP-1 cells infection with VSV, and it was observed that *cdk2* was not upregulated. Fish *cdk2* was significantly increased upon infection with fish viruses CyHV-2 or SVCV in *Carassius auratus gibelio*, *Danio rerio*, and *Pimephales promelas*, compared to human *cdk2*, indicating that fish CDK2 is involved in the host's antiviral response. Meanwhile, specific qPCR and

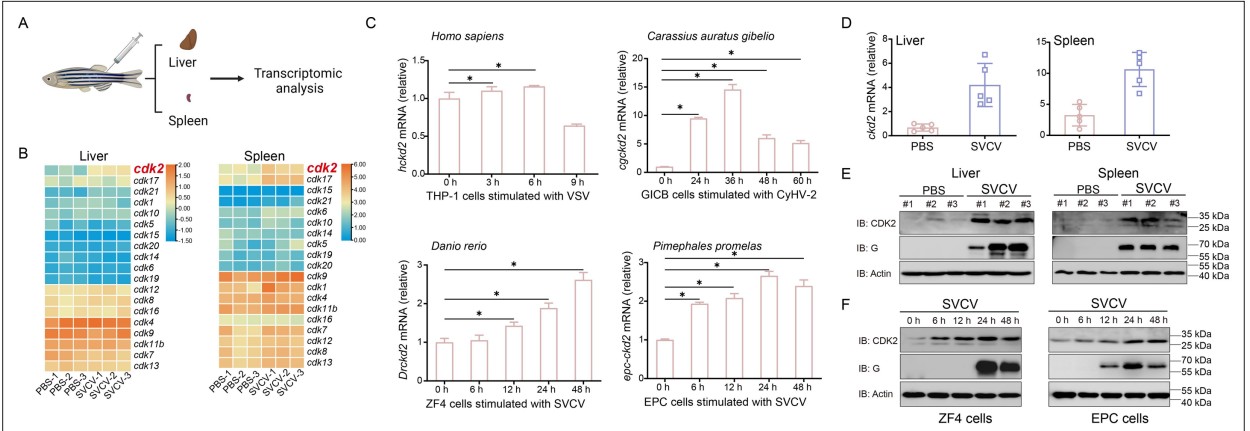

**Figure 1.** In vivo and in vitro, CDK2 is upregulated during viral infection. (**A**) Schematic representation of zebrafish tissue dissection and RNA extraction for transcriptome sequencing. The liver and spleen tissues from male and female zebrafish injected with phosphate-buffered saline (PBS) (5 μL/individual), spring viremia of carp virus (SVCV) ($5\times10^8$ $TCID_{50}$/mL, 5 μL/individual) for 48 hr. Total RNAs were extracted and used for transcriptome sequencing and analysis. (**B**) Heatmap view of mRNA variations of CDKs in the liver and spleen of zebrafish with or without SVCV infection. (**C**) qPCR analysis of *cdk2* mRNA in the THP-1, GICB, ZF4, or epithelioma papulosum cyprini (EPC) cells infected with vesicular stomatitis virus (VSV), CyHV-2, or SVCV for the indicated times. Representative experiments are shown (n=3). (**D**) qPCR analysis of *cdk2* mRNA in the liver and spleen of zebrafish (n=5 per group) given injected intraperitoneally (i.p.) with PBS or SVCV for 48 hr. (**E**) IB of proteins in the liver and spleen of zebrafish (n=3 per group) given i.p. injection of PBS or SVCV for 48 hr. (**F**) IB of proteins in ZF4 and EPC cells infected with SVCV for the indicated times. Representative experiments are shown (n=3).

The online version of this article includes the following source data for figure 1:

**Source data 1.** PDF file containing original western blots for *Figure 1E and F*, indicating the relevant bands and treatments.

**Source data 2.** Original files for western blot analysis displayed in *Figure 1E and F*.

**Source data 3.** Original data for graphs analysis in *Figure 1B and C*.

immunoblot analysis to validate this finding in zebrafish tissues, which demonstrated that the *cdk2* transcript level was significantly increased upon SVCV infection. The CDK2 protein level was also consistently higher in the viral infection group (*Figure 1D and E*). We investigated whether this CDK2 expression pattern also existed in vitro. For this purpose, we used the zebrafish embryonic fibroblast cell line (ZF4) and another cyprinid fish cell line (epithelioma papulosum cyprini (EPC)). During the 48 hr (h) viral infection period, CDK2 was significantly upregulated at the protein level (*Figure 1F*). Taken together, the in vivo and in vitro data suggest that fish CDK2 may play a role in the response to viral infection.

## *cdk2*−/− fish exhibit effective antiviral capacities

To investigate the role of CDK2 in the antiviral process, we generated a *cdk2* knockout zebrafish homozygote (*cdk2*−/−). Our findings demonstrated that the survival rate under viral infection was significantly higher in the *cdk2*−/− group than in the wild-type group (*Figure 2A*). We selected the liver, spleen, and kidney as representative tissues in which to analyze the *cdk2*−/− antiviral capacity. In the H&E staining assay, severe tissue damage was observed in the wild-type group, while the *cdk2*−/− group displayed dramatically less tissue damage (*Figure 2B*). The viral transcripts in these tissues were observed and then compared to the abundance in the wild-type. The replication of viral genes, such as SVCV N, was typically inhibited in *cdk2*−/− homozygote tissues (*Figure 2C*). Viral protein level analysis also confirmed these results, as SVCV G, N, and P proteins were suppressed in *cdk2*−/− homozygotes (*Figure 2D*).

Transcriptomics analysis was employed to investigate the antiviral regulation by CDK2. Differential expression analysis identified 1707 upregulated genes and 1490 downregulated genes in the liver of *cdk2* knockout zebrafish versus WT zebrafish after SVCV infection (*Figure 2E*). Moreover, gene set enrichment analyses (GSEAs) were performed to analyze the IFN response to virus infection-related genes, which demonstrated that these genes were significantly activated in the *cdk2*−/− group (*Figure 2F*). Differential expression analysis further showed that *cdk2* knockout results in the upregulation of many IFN-stimulated genes (ISGs) after SVCV infection (*Figure 2G*). To procure the transcriptome data, IFN was assayed in zebrafish tissues, including liver, spleen, and kidney. Similar to

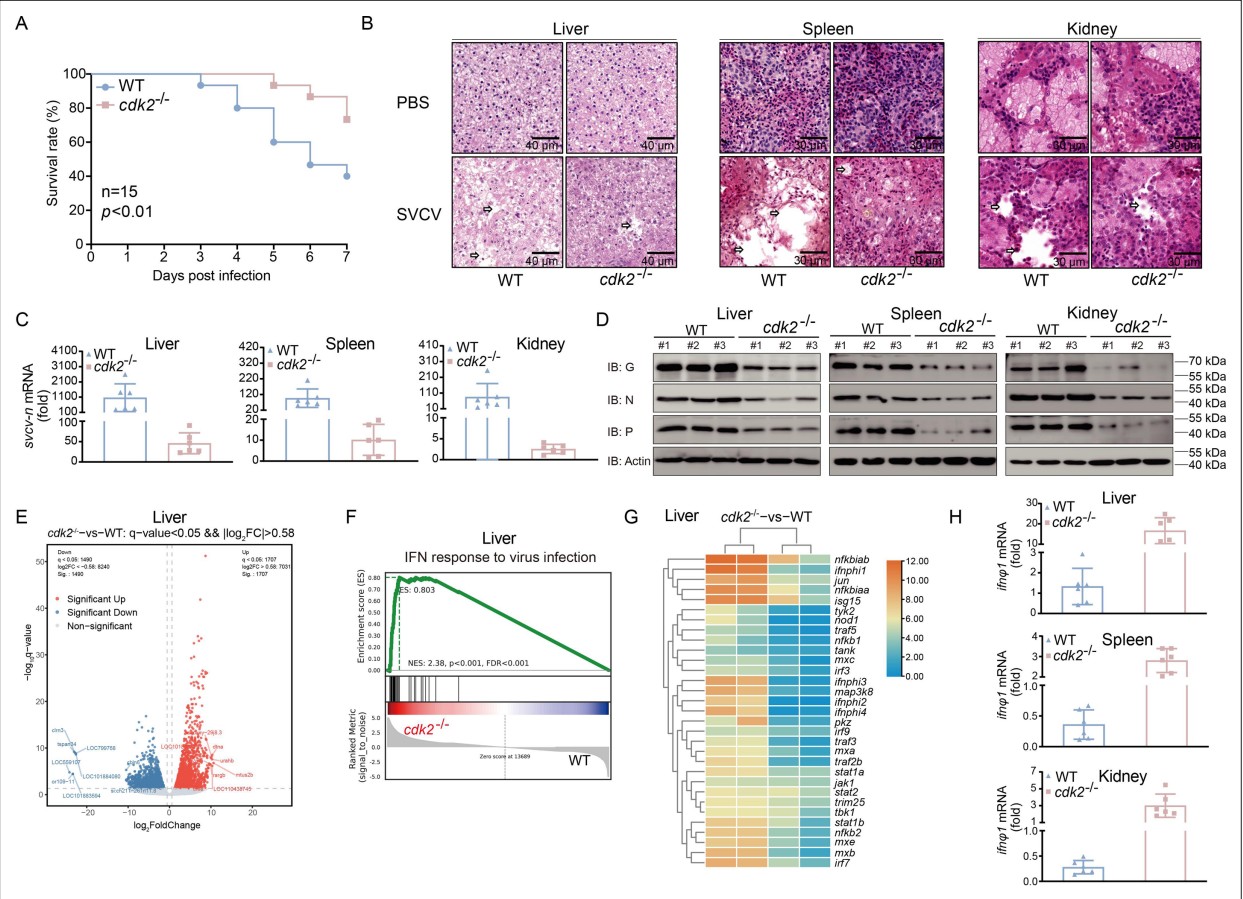

**Figure 2.** CDK2 deficiency protects fish from spring viremia of carp virus (SVCV) infection. (**A**) Survival (Kaplan-Meier Curve) of *cdk2*[+/+] and *cdk2*[-/-] zebrafish (n=15 per group) at various days after i.p. infected with SVCV (5×10$^8$ TCID$_{50}$/mL, 5 μL/individual). (**B**) Microscopy of hematoxylin and eosin (H&E)-stained liver, spleen, and kidney sections from *cdk2*[+/+] and *cdk2*[-/-] zebrafish treated with SVCV for 48 hr. (**C**) qPCR analysis of *svcv-n* mRNA in the liver, spleen, and kidney of *cdk2*[+/+] and *cdk2*[-/-] zebrafish (n=6 per group) given i.p. injection of SVCV for 48 hr. (**D**) Immunoblotting (IB) of proteins in the liver, spleen, and kidney sections from *cdk2*[+/+] and *cdk2*[-/-] zebrafish (n=3 per group) treated with SVCV for 48 hr. (**E**) CDK2 regulates antiviral response-relevant target genes, presented as a volcano plot of geneiss with differential expression after SVCV infection in the liver of *cdk2*[+/+] and *cdk2*[-/-] zebrafish. (**F**) Gene set enrichment analysis (GSEA) of differentially expressed genes in the liver of *cdk2*[+/+] and *cdk2*[-/-] zebrafish with SVCV infection and enrichment of interferon (IFN). FDR (*q*-value) was shown. (**G**) Heatmap view of mRNA variations of IFN-mediated IFN-stimulated gene (ISG) sets in the liver of *cdk2*[+/+] and *cdk2*[-/-] zebrafish with SVCV infection. (**H**) qPCR analysis of *ifnφ1* mRNA in the liver, spleen, and kidney of *cdk2*[+/+] and *cdk2*[-/-] zebrafish (n=6 per group) given i.p. injection of SVCV for 48 hr.

The online version of this article includes the following source data for figure 2:

**Source data 1.** PDF file containing original western blots for *Figure 2D*, indicating the relevant bands and treatments.

**Source data 2.** Original files for western blot analysis displayed in *Figure 2D*.

**Source data 3.** Original data for graphs analysis in *Figure 2A, C, G and H*.

the transcriptome assay, IFN was found to be significantly higher in the *cdk2*[-/-] groups, indicating that stronger antiviral capacity and higher IFN transcription occurred in *cdk2*[-/-] fish (*Figure 2H*). Collectively, these data suggest that viral replication was hindered in the *cdk2*[-/-] fish.

## CDK2 inhibits IFN expression and promotes viral proliferation

The impact of CDK2 on viral infection was examined in relation to IFN. The IFN response is an essential mechanism for host resistance against viruses. Overexpression of CDK2 decreased the IFN promoter and ISRE motif activation by SVCV or poly I:C (*Figure 3A*). An effective shcdk2 was produced and identified (*Figure 3B*). The impact of CDK2 knockdown was explored, with the results suggesting that the downregulation of CDK2 facilitates IFN promoter activity (*Figure 3C*). The mRNA levels of *ifn* and downstream ISG *vig1* transcription were monitored, revealing that CDK2 caused a significant

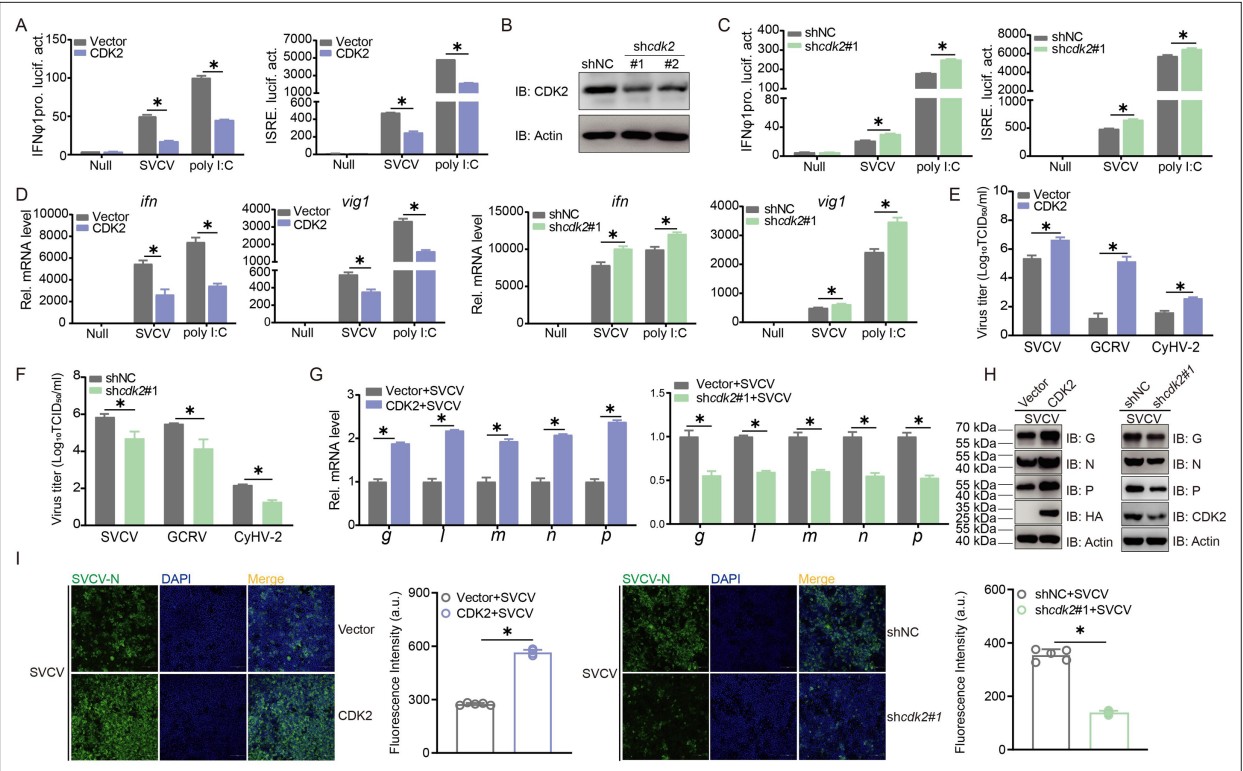

**Figure 3.** CDK2 negatively regulates interferon (IFN) production and promotes viral replication. (**A and C**) Luciferase activity of IFNφ1pro and ISRE in epithelioma papulosum cyprini (EPC) cells transfected with indicated plasmids for 24 hr, and then untreated or infected with spring viremia of carp virus (SVCV) (MOI = 1) or transfected with poly I:C (0.5 µg) for 24 hr before luciferase assays. (**B**) IB of proteins in EPC cells transfected with indicated plasmids for 24 hr. (**D**) qPCR analysis of *ifn* and *vig1* in EPC cells transfected with indicated plasmids for 24 hr, and then untreated or infected with SVCV (MOI = 1) or transfected with poly I:C (2 µg) for 24 hr. (**E and F**) Plaque assay of virus titers in EPC, *Ctenopharyngodon idellus* kidney (CIK), and Gibel carp brain (GiCB) cells transfected with indicated plasmids for 24 hr, followed by SVCV, Grass carp reovirus (GCRV), and CyHV-2 challenge for 24–72 hr. (**G and H**) qPCR and immunoblotting (IB) analysis of SVCV genes in epithelioma papulosum cyprini (EPC) cells transfected with indicated plasmids for 24 hr, followed by SVCV challenge for 24 hr. (**I**) Interferon (IF) analysis of N protein in EPC cells transfected with indicated plasmids for 24 hr, followed by SVCV challenge for 24 hr. The fluorescence intensity (arbitrary unit, a.u.) was recorded by the LAS X software, and the data were expressed as mean ± SD, n=5. All experiments were repeated for at least three times with similar results.

The online version of this article includes the following source data for figure 3:

**Source data 1.** PDF file containing original western blots for **Figure 3B and H** indicating the relevant bands and treatments.

**Source data 2.** Original files for western blot analysis displayed in **Figure 3B and H**.

**Source data 3.** Original data for graphs analysis in **Figure 3A, C–G and I**.

decrease in *ifn* and *vig1*, whereas the knockdown of CDK2 increased the IFN response (**Figure 3D**). In the antiviral capacity assays, CDK2 from zebrafish, gibel carp, and grass carp all promoted the proliferation of their respective corresponding viruses (**Figure 3E**). Conversely, CDK2 knockdown significantly suppressed viral proliferation (**Figure 3F**). Cells overexpressing CDK2 showed enhanced production of viral mRNA and proteins, while CDK2-knockdown cells showed attenuation of the same viral mRNA and proteins (**Figure 3G and H**). Immunofluorescence revealed a higher intensity of green signals indicating SVCV N protein in the CDK2 overexpression group compared to the control, while a lower green signal was observed in the CDK2-knockdown group compared to the normal group (**Figure 3I**). These findings suggest that CDK2 inhibits IFN expression in the host and reduces antiviral capacity.

## CDK2 interacts with TBK1 and reduces its expression

The RLR pathway plays an important role in IFN activation. To investigate whether CDK2 counters RLR signaling, as observed earlier, we examined whether CDK2 inhibits RLR factors that activate IFN promoters. The activation of the IFN promoters induced by MAVS and TBK1 was significantly inhibited by CDK2, whereas it was unaffected by MITA (**Figure 4A**). CDK2 knockdown restored MAVS

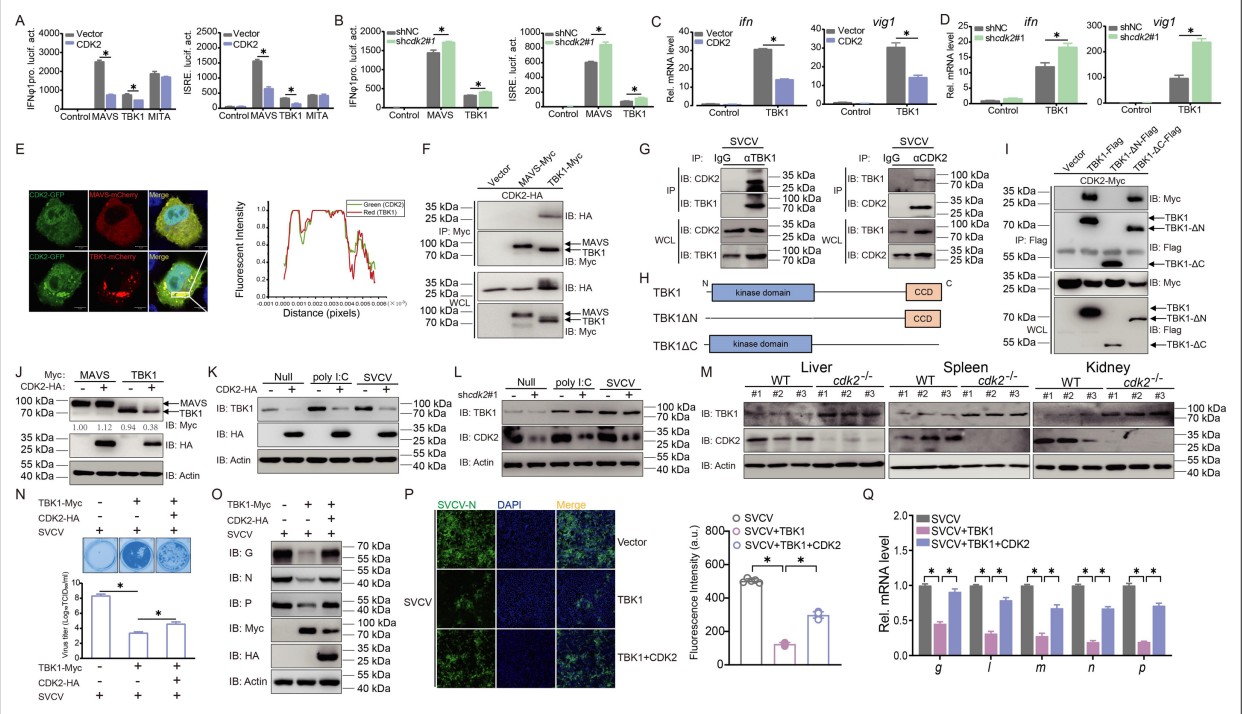

**Figure 4.** CDK2 associates with TBK1 and mediates its degradation. (**A and B**) Luciferase activity of IFNφ1pro and ISRE in epithelioma papulosum cyprini (EPC) cells transfected with indicated plasmids for 24 hr. (**C and D**) qPCR analysis of *ifn* and *vig1* in EPC cells transfected with indicated plasmids for 24 hr. (**E**) Confocal microscopy of CDK2 and TBK1 in EPC cells transfected with indicated plasmids for 24 hr. The coefficient of colocalization was determined by qualitative analysis of the fluorescence intensity of the selected area in Merge. (**F**) Immunoblotting (IB) of whole cell lysates (WCLs) and proteins immunoprecipitated with anti-Myc antibody-conjugated agarose beads from EPC cells transfected with indicated plasmids for 24 hr. (**G**) IB of WCLs and proteins immunoprecipitated with anti-TBK1 or anti-CDK2 antibody from EPC cells infected with spring viremia of carp virus (SVCV) for 24 hr. (**A–G**) Representative experiments are shown (n=3). (**H**) Schematic representation of full-length TBK1 and its mutants. (**I**) IB of WCLs and proteins immunoprecipitated with anti-Flag antibody-conjugated agarose beads from EPC cells transfected with indicated plasmids for 24 hr. (**J**) IB of proteins in EPC cells transfected with indicated plasmids for 24 hr. (**K and L**) IB of proteins in EPC cells transfected with CDK2-HA or sh*cdk2*#1 for 24 hr, followed by untreated or infected with SVCV (MOI = 1) or transfected with poly I:C (2 μg) for 24 hr. (**I–L**) Representative experiments are shown (n=3). (**M**) IB of proteins in the liver, spleen, and kidney sections from *cdk2*[+/+] and *cdk2*[-/-] zebrafish (n=3 per group). (**N**) Plaque assay of virus titers in EPC cells transfected with indicated plasmids for 24 hr, followed by SVCV challenge for 24–48 hr. (**O and Q**) IB and qPCR analysis of SVCV genes in EPC cells transfected with indicated plasmids for 24 hr, followed by SVCV challenge for 24 hr. (**P**) Interferon (IF) analysis of N protein in EPC cells transfected with indicated plasmids for 24 hr, followed by SVCV challenge for 24 hr. The fluorescence intensity (arbitrary unit, a.u.) was recorded by the LAS X software, and the data were expressed as mean ± SD, n=5. (**N–P**) Representative experiments are shown (n=3).

The online version of this article includes the following source data for figure 4:

**Source data 1.** PDF file containing original western blots for *Figure 4F-G, I–M and O* indicating the relevant bands and treatments.

**Source data 2.** Original files for western blot analysis displayed in *Figure 4F-G, I–M and O*.

**Source data 3.** Original data for graphs analysis in *Figure 4A-E, N and P–Q*.

and TBK1 functions following IFN promoter stimulation (*Figure 4B*). Moreover, the *ifn* and *vig1* transcripts were monitored. The results showed that CDK2 notably reduced *ifn* and *vig1* mRNA levels, whereas the CDK2 knockdown improved the IFN response (*Figure 4C and D*). These findings suggest that CDK2 targets either MAVS or TBK1. The subcellular localization of CDK2 was observed to be distributed throughout the cell. However, TBK1 and MAVS exhibited cytoplasmic localization, while there appeared to be a significant punctate overlap between CDK2 and TBK1, indicating a possible association between CDK2 and TBK1 (*Figure 4E*). Subsequently, association analysis of CDK2 and MAVS or TBK1 was conducted. The co-IP assay revealed an interaction between CDK2 and TBK1, yet not between CDK2 and MAVS (*Figure 4F*). Afterward, an endogenous co-IP assay was performed to confirm the interaction between CDK2 and TBK1, while the interaction was also verified with TBK1 or CDK2 being enriched (*Figure 4G*). To identify the essential domain that mediates the interaction with CDK2, two truncated TBK1 mutants were generated (*Figure 4H*). The TBK1-ΔN-flag, which lacked

the kinase domain, meaning it cannot bind to CDK2, indicated that the kinase domain in TBK1 is necessary for it to associate with CDK2 (*Figure 4I*). Since the interaction between CDK2 and TBK1 was confirmed, whether CDK2 affects TBK1 stability was also investigated. Co-overexpression of the relevant RLR factors and CDK2 demonstrated that TBK1 expression was substantially decreased in the presence of CDK2, compared to MAVS (*Figure 4J*). Consistently, CDK2 impaired endogenous TBK1 under both normal and stimulation states, whereas CDK2 knockdown abolished this effect, indicating that CDK2 impaired TBK1 expression (*Figure 4K and L*). The expression of TBK1 was also significantly increased in the liver, spleen, and kidney of *cdk2*⁻/⁻ fish, confirming the in vivo impact of CDK2 on TBK1 expression (*Figure 4M*). Subsequently, its biological function was studied to investigate whether CDK2 affects the crucial antiviral role of TBK1. During SVCV infection, cells overexpressing TBK1 showed little CPE; however, CDK2 dramatically counteracted the antiviral capacity of TBK1, as confirmed by virus titer identification (*Figure 4N*). For viral protein expression, CDK2 restored the TBK1-mediated prevention of viral protein expression, including SVCV G, N, and P proteins. Immuno-fluorescence also demonstrated that the inhibition of SVCV N protein by TBK1 was restored following CDK2 overexpression (*Figure 4O and P*). CDK2 also restored viral nucleic acid transcription, which was abolished by TBK1 (*Figure 4Q*). These results demonstrate that CDK2 interacts with the kinase domain in TBK1, leading to reduced expression and a weakened antiviral effect.

## CDK2 causes TBK1 degradation by increasing K48-linked polyubiquitination

The next step was to investigate the mechanisms through which CDK2 negatively regulates TBK1. To determine whether the decrease in TBK1 occurred at the mRNA or protein level, *tbk1* transcription was monitored. CDK2 had little effect on *tbk1* in either the control or virus-infected groups (*Figure 5A*). Therefore, attention was placed on modulating TBK1 at the protein level. Various reagents that inhibit the ubiquitin (Ub)-proteasome and autophagosome, such as MG132, 3-MA, Baf-A1, and CQ, were utilized to clarify the precise regulatory mechanism. Compared to the autophagosome inhibitors 3-MA, Baf-A1, and CQ, using the Ub-proteasome inhibitor MG132 significantly impeded TBK1 degradation in a dose-dependent manner (*Figure 5B and C*). This suggests that the degradation of TBK1 by CDK2 is proteasome-dependent. Similar results were observed for endogenous TBK1 in the presence or absence of poly I:C stimulation or SVCV infection (*Figure 5D*). Since ubiquitination is an important process during proteasome-dependent degradation, we next investigated whether ubiquitination was important in the CDK2-mediated degradation of TBK1. Expectedly, immunoblot analysis confirmed that CDK2 increased the ubiquitination of TBK1 (*Figure 5E*). Polyubiquitin chain modification, either K48-linked or K63-linked, can either target proteins for degradation or increase stability. To investigate how TBK1 was modified, we performed MS analysis of ubiquitinated TBK1 from cells and found that K48-linked polyubiquitinated TBK1 was readily detected in CDK2-overexpressed cells, whereas K63-linked polyubiquitinated TBK1 was hardly detected (*Figure 5F*). Consistently, when we transfected EPC cells with TBK1-Myc, CDK2-HA, WT-Ub, K48-Ub, or K63-Ub, we found that CDK2 markedly increased K48-linked, but not K63-linked, ubiquitination of TBK1, while CDK2 knockdown remarkably attenuated K48-linked ubiquitination of TBK1 (*Figure 5G and H*). Taken together, these findings indicate that the CDK2 triggers the degradation of TBK1 through K48-linked ubiquitination.

## CDK2 recruits the ubiquitin E3 ligase Dtx4 to interact with and degrade TBK1

Since CDK2 is not an E3 ubiquitin ligase, it is speculated that CDK2-mediated degradation of TBK1 does not occur directly; instead, CDK2 acts as an adaptor to recruit an E3 ubiquitin ligase to TBK1. Several homologs of TBK1-associated E3 ubiquitin ligases in mammals were cloned into fish to detect protein interactions. It was found that Trim11 and Dtx4 are associated with TBK1, and these interactions were verified through inverse experiments (*Figure 6A and B*). Moreover, protein interactions between CDK2 and Trim11 or Dtx4 were investigated. Dtx4 exhibited a significant interaction with CDK2 but not with Trim11, indicating that Dtx4 interacts with both TBK1 and CDK2 (*Figure 6C*). Hence, Dtx4 was the main focus of the subsequent assays. Since Dtx4 is a potential ubiquitin ligase for TBK1, we verified whether CDK2 facilitated the interaction between Dtx4 and TBK1. Overexpression of CDK2 significantly increased the interaction between Dtx4 and TBK1, while CDK2 knockdown impeded this progress, suggesting that CDK2 promotes the interaction between Dtx4 and TBK1 (*Figure 6D*). Next,

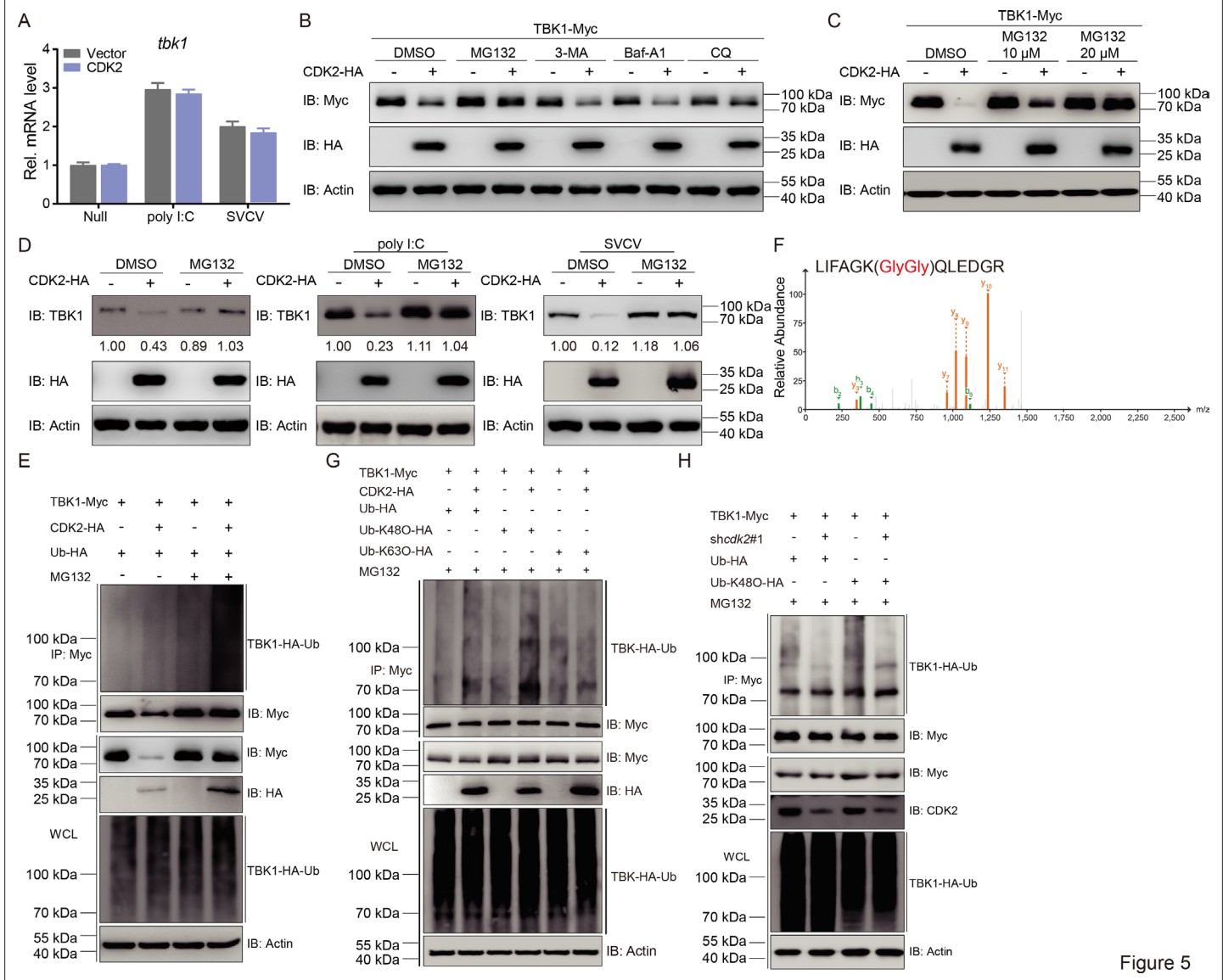

**Figure 5.** CDK2 increases the K48-linked polyubiquitination of TBK1. (**A**) qPCR analysis of *epc-tbk1* in epithelioma papulosum cyprini (EPC) cells transfected with indicated plasmids for 24 hr, and then untreated or infected with spring viremia of carp virus (SVCV) (MOI = 1) or transfected with poly I:C (2 µg) for 24 hr. (**B and C**) Immunoblotting (IB) of proteins in EPC cells transfected with indicated plasmids for 18 hr, followed by treatments of MG132 (10 µM), 3-MA (2 mM), Baf-A1 (100 nM), and CQ (50 µM) for 6 hr, respectively. (**D**) IB of proteins in EPC cells transfected with CDK2-HA for 24 hr, followed by untreated or infected with SVCV (MOI = 1) or transfected with poly I:C (2 µg) for 24 hr. Protein lysates were harvested after MG132 (20 µM) treatments (6 hr) for IB analysis. (**E**) TBK1 ubiquitination assays in EPC cells transfected with indicated plasmids for 18 hr, followed by DMSO or MG132 treatments for 6 hr. (**A–E**) Representative experiments are shown (n=3). (**F**) Mass spectrometry analysis of a peptide derived from ubiquitinated TBK1-Myc. (**G and H**) TBK1 ubiquitination assays in EPC cells transfected with indicated plasmids for 18 hr, followed by MG132 treatments for 6 hr. Representative experiments are shown (n=3).

The online version of this article includes the following source data for figure 5:

**Source data 1.** PDF file containing original western blots for *Figure 5B-E and G–H* indicating the relevant bands and treatments.

**Source data 2.** Original files for western blot analysis displayed in *Figure 5B–E and G–H*.

**Source data 3.** Original data for graphs analysis in *Figure 5A*.

the impact of the Dtx4 interaction was elucidated. Overexpression of DTX4 amplified CDK2-mediated inhibition of IFN promoter activity induced by TBK1 (*Figure 6E*). Moreover, CDK2-mediated reduction of *ifn* and *vig1* mRNAs activated by TBK1 was also augmented by Dtx4 overexpression (*Figure 6F*). In contrast to the normal state where CDK2 degrades TBK1, Dtx4 overexpression increased degradation

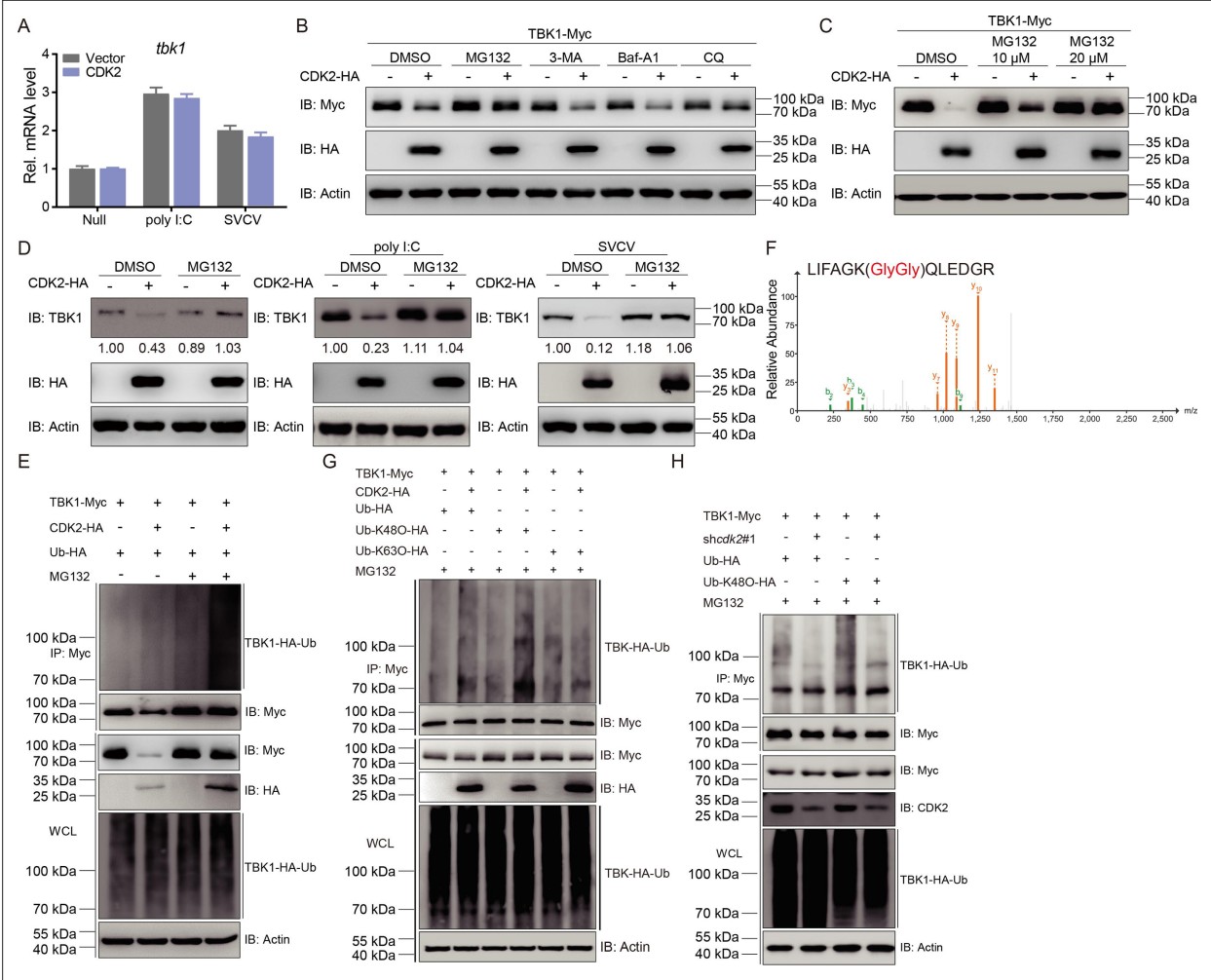

**Figure 6.** CDK2 recruits Dtx4 to degrade TBK1. (**A–D**) Immunoblotting (IB) of whole cell lysates (WCLs) and proteins immunoprecipitated with anti-Myc or Flag Ab-conjugated agarose beads from epithelioma papulosum cyprini (EPC) cells transfected with indicated plasmids for 24 hr. (**E and J**) Luciferase activity of IFNφ1pro in EPC cells transfected with indicated plasmids for 24 hr. (**F and K**) qPCR analysis of *ifn* and *vig1* in EPC cells transfected with indicated plasmids for 24 hr. (**G and L**) IB of proteins in EPC cells transfected with indicated plasmids for 24 hr. (**H and M**) IB of proteins in EPC cells transfected with CDK2-HA and Dtx4-Myc or sh*dtx4*#1 for 24 hr, followed by untreated or infected with spring viremia of carp virus (SVCV) (MOI = 1) or transfected with poly I:C (2 µg) for 24 hr. All experiments were repeated for at least three times with similar results.

The online version of this article includes the following source data for figure 6:

**Source data 1.** PDF file containing original western blots for *Figure 6A-D, G–I and L–M* indicating the relevant bands and treatments.

**Source data 2.** Original files for western blot analysis displayed in *Figure 6A-D, G–I and L–M*.

**Source data 3.** Original data for graphs analysis in *Figure 6E-F and J–K*.

(*Figure 6G*). The degradation of endogenous TBK1 by CDK2 was more severe when Dtx4 was over-expressed (*Figure 6H*). Two shRNAs were designed and generated, and after validation of exogenous and endogenous knockdown efficiencies, sh*dtx4*#1 was selected for the following assay (*Figure 6I*). Dtx4 knockdown remarkably abrogated CDK2 suppression of TBK1-induced IFN promoter activity and *ifn* and *vig1* mRNAs, and CDK2 regulated both the exogenous and endogenous degradation of TBK1 (*Figure 6J–M*). Overall, CDK2 recruits the E3 ubiquitin ligase Dtx4 to degrade TBK1.

## The K567 site in TBK1 plays an essential role in CDK2-mediated degradation

By accelerating CDK2-mediated TBK1 degradation, the precise ubiquitin ligase function of Dtx4 was identified. Compared to Trim11, Dtx4 significantly enhanced the CDK2-potentiated ubiquitination

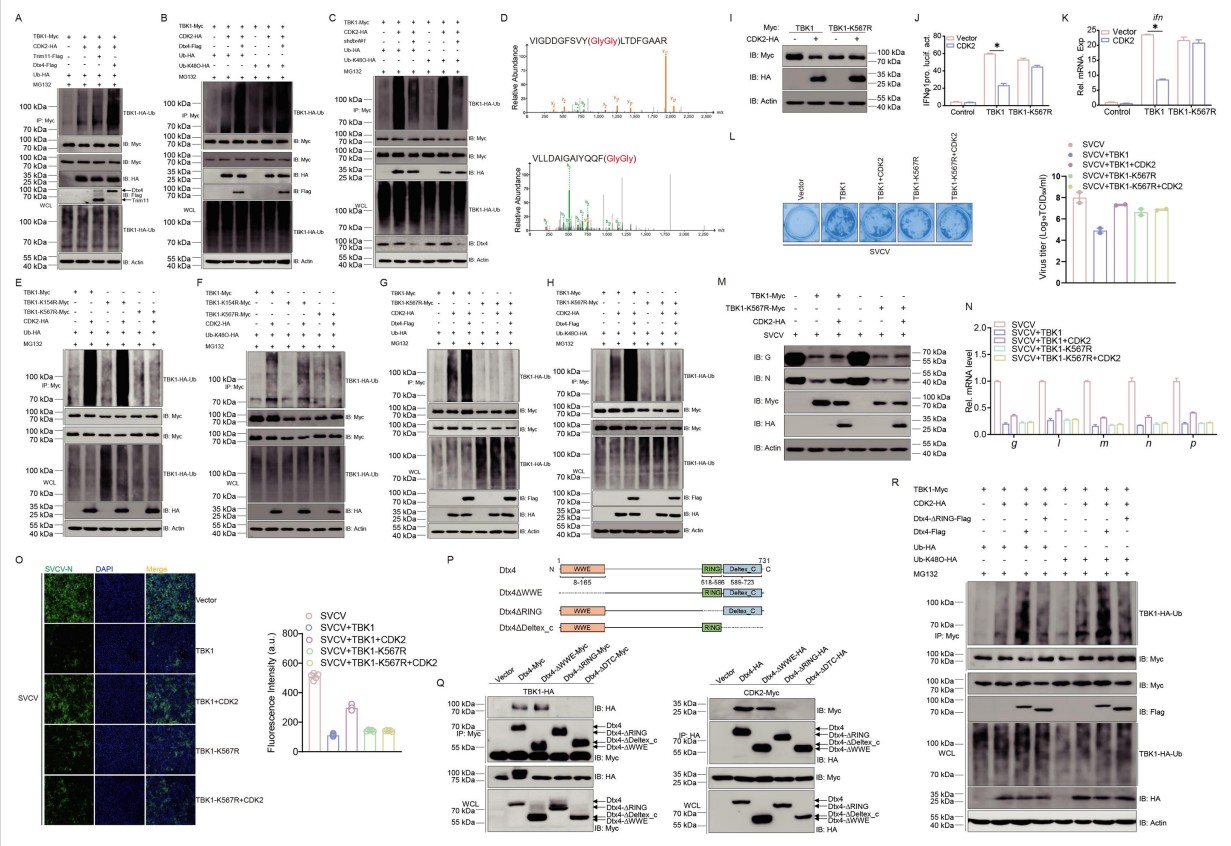

**Figure 7.** K567 site is critical for the ubiquitination degradation of TBK1. (**A–C**) TBK1 ubiquitination assays in epithelioma papulosum cyprini (EPC) cells transfected with indicated plasmids for 18 hr, followed by MG132 treatments for 6 hr. Representative experiments are shown (n=3). (**D**) Mass spectrometry analysis to show that K154 and K567 of TBK1 is conjugated with ubiquitin. (**E–H**) TBK1 ubiquitination assays in EPC cells transfected with indicated plasmids for 18 hr, followed by MG132 treatments for 6 hr. (**I**) Immunoblotting (IB) of proteins in EPC cells transfected with indicated plasmids for 24 hr. (**J**) Luciferase activity of IFNφ1pro in EPC cells transfected with indicated plasmids for 24 hr. (**K**) qPCR analysis of *ifn* in EPC cells transfected with indicated plasmids for 24 hr. (**L**) Plaque assay of virus titers in EPC cells transfected with indicated plasmids for 24 hr, followed by spring viremia of carp virus (SVCV) challenge for 24–48 hr. (**M and N**) qPCR and IB analysis of SVCV genes in EPC cells transfected with indicated plasmids for 24 hr, followed by SVCV challenge for 24 hr. (**O**) Interferon (IF) analysis of N protein in EPC cells transfected with indicated plasmids for 24 hr, followed by SVCV challenge for 24 hr. The fluorescence intensity (arbitrary unit, a.u.) was recorded by the LAS X software, and the data were expressed as mean ± SD, n=5. (**E–O**) Representative experiments are shown (n=3). (**P**) Schematic representation of full-length Dtx4 and its mutants. (**Q**) IB of whole cell lysates (WCLs) and proteins immunoprecipitated with anti-Myc or HA Ab-conjugated agarose beads from EPC cells transfected with indicated plasmids for 24 hr. (**R**) TBK1 ubiquitination assays in EPC cells transfected with indicated plasmids for 18 hr, followed by MG132 treatments for 6 hr. (**Q and R**) Representative experiments are shown (n=3).

The online version of this article includes the following source data for figure 7:

**Source data 1.** PDF file containing original western blots for *Figure 7A-C, E–I, M and Q–R* indicating the relevant bands and treatments.

**Source data 2.** Original files for western blot analysis displayed in *Figure 7A-C, E–I, M and Q–R*.

**Source data 3.** Original data for graphs analysis in *Figure 7J-L and N-O*.

of TBK1 (*Figure 7A*). Further analysis revealed that Dtx4 overexpression augmented K48-linked TBK1 ubiquitination and that Dtx4 knockdown reduced this ubiquitination. Thus, demonstrating that Dtx4 is critical for CDK2-mediated TBK1 ubiquitination (*Figure 7B and C*). To analyze the molecular mechanism of TBK1 modification by ubiquitination, two potential TBK1 lysine sites, namely K154 and K567, were identified for modification by ubiquitination using mass spectrometry analysis (*Figure 7D*). Subsequently, two TBK1 mutants were generated with point mutations by mutating K154 and K567 to R to create K154R and K567R. In the ubiquitination assay, TBK1-K567R could not be ubiquitinated by CDK2, while K154R was almost identical to the wild-type TBK1 (*Figure 7E*). This result was also characterized in the K48-linked ubiquitination assay (*Figure 7F*). Moreover, Dtx4 significantly increased CDK2-mediated TBK1 wild-type and K48-linked ubiquitination but failed to enhance these

processes in the K567R mutant group of TBK1 (*Figure 7G and H*). This strongly suggests that K567 is a crucial site for TBK1 ubiquitination by Dtx4. Additionally, when TBK1-K567 was mutated, CDK2-mediated degradation of TBK1 failed, and the IFN induction and antiviral capacity restriction of TBK1 by CDK2 was also ineffective (*Figure 7I–O*). These findings confirm the essential role of TBK1-K567 in recruiting Dtx4 through CDK2. The RING domain is crucial for E3 ubiquitin ligase activity. Three truncated mutants of Dtx4 were generated (*Figure 7P*). The mutant lacking the RING domain (Dtx4-△RING) significantly impaired the interaction between Dtx4 and TBK1 or CDK2 (*Figure 7Q*). In the ubiquitination assay, the wild-type Dtx4 enhanced TBK1 and K48-linked ubiquitination. On the other hand, Dtx4-△RING failed, indicating that the RING domain was necessary for TBK1 ubiquitination by Dtx4 (*Figure 7R*). In summary, the data above demonstrates that the K567 site in TBK1 and the RING domain in the E3 ubiquitin ligase Dtx4 are crucial for CDK2 ubiquitination, leading to the degradation of TBK1.

## Discussion

Although IFN responses exhibit powerful functions in defending against viral infection, excessive activation of IFN production may cause autoimmune disease. Therefore, the host needs to develop a set of regulatory mechanisms to balance immune responses. In this study, we illustrated a novel role for fish CDK2 in the negative regulation of IFN expression, which degraded TBK1 for K48-linked ubiquitination by recruiting E3 ubiquitin ligase Dtx4.

TBK1 is a central kinase in MAVS, STING, and TIR-domain-containing adapter-inducing IFN-β (TRIF) signaling complexes, which promote the phosphorylation of IRF3 and the production of IFN. Thus, the activation of TBK1 must be tightly regulated to avoid excessive autoimmune responses. TBK1 activity can be regulated in various ways, including phosphorylation, ubiquitination, SUMOylation, and preventing the formation of functional TBK1-containing complexes. For instance, receptor tyrosine kinase HER2 recruits AKT1 to directly phosphorylate TBK1, which disrupts the TBK1–STING association and K63-linked ubiquitination of TBK1, thus suppressing antiviral responses (*Wu et al., 2019*). Siglec1 associates with DAP12 and SHP2 to recruit the E3 ubiquitin ligase TRIM27, which induces K48-linked-ubiquitination-mediated TBK1 degradation, resulting in the inhibition of IFN production (*Zheng et al., 2015*). Similar to mammals, TBK1 also dramatically activates the IFN signaling pathway in fish. Therefore, its activation must be precisely modulated. For example, cytokine receptor-like factor 3 (Crlf3) promotes the degradation of TBK1 via K48-linked ubiquitination, resulting in inhibition of IFN production (*Yan et al., 2023*). While TBK1 regulation has received attention, multiple molecules and underlying molecular mechanisms have not been fully characterized as potential TBK1 targets.

Ubiquitination is one of the most versatile post-translational modifications (PTMs) of proteins and plays numerous vital roles in regulating antiviral responses. Ubiquitin comprises seven lysine residues (K6, K11, K27, K29, K33, K48, and K63); thus, seven different polyubiquitin chains can be produced (*Song and Luo, 2019*). K48-linked polyubiquitination is used to signal for proteasomal degradation of substrate proteins. In contrast, K63-linked polyubiquitination is a non-proteolytic mode of modification that plays several vital roles in stabilizing and activating target proteins (*Kulathu and Komander, 2012*). Multiple studies have shown that different polyubiquitin chains can regulate the expression and activation of TBK1 (*Zhao, 2013*). For instance, Nrdp1, MIB1/2, RNF128, and NLRC4 mediate the K63-linked polyubiquitination of TBK1 and facilitate its activation (*Wang et al., 2009*; *Ye et al., 2014*; *Song et al., 2016*; *Zhang et al., 2023*). TRIP, SOCS3, and TRAF3IP3 have been proven to negatively regulate the IFN signaling pathway by targeting TBK1 for K48-linked polyubiquitination and degradation (*Zhang et al., 2012*; *Liu et al., 2015*; *Deng et al., 2020*). Ubiquitination modification is a tightly regulated and reversible process that maintains cellular homeostasis. For example, the deubiquitinating enzymes CYLD, RNF114B, USP2b, and UBE2S remove the K63-linked polyubiquitination of TBK1 to block its activation (*Friedman et al., 2008*; *Zhang et al., 2019*; *Zhang et al., 2014*; *Huang et al., 2020*). Additionally, the USP1–UAF1 deubiquitinate complex has been found to cleave the K48-linked polyubiquitination of TBK1 to reverse its degradation process (*Yu et al., 2017*). These findings are specific to the ubiquitination of mammalian TBK1. Although fish TBK1 is highly conserved compared to mammalian TBK1, its ubiquitination modifications are worth exploring. Our study identified a previously unrecognized role for CDK2 in promoting the K48-linked polyubiquitination and proteasomal degradation of fish TBK1.

A series of ubiquitin-related enzymes are responsible for ubiquitination, including ubiquitin-activating enzymes (E1s), ubiquitin-conjugating enzymes (E2s), and ubiquitin ligases (E3s). Among them, the E3s are the critical components that determine the substrate specificity (*Pickart, 2001*). E3s are generally divided into two large classes, including the homology to the E6-associated protein carboxyl terminus (HECT) domain-containing E3 ligases and the really interesting new gene (RING) domain-containing E3 ligases (*Metzger et al., 2012*). Multiple studies have demonstrated that E3 ligases, including TRIP, Socs3, Dtx4, and TRIM27, specifically target TBK1 for K48-linked polyubiquitination and degradation (*Zheng et al., 2015*; *Zhang et al., 2012*; *Liu et al., 2015*; *Cui et al., 2012*). However, CDK2 is not an E3 ubiquitin ligase. Thus, we reasoned that CDK2 might be a mediator in the recruitment of an E3 ubiquitin ligase for K48-linked polyubiquitination. To validate this hypothesis, we investigated and screened hTBK1-associated E3 ubiquitin ligases, demonstrating that Dtx4 interacted with CDK2 and enhanced the K48-linked polyubiquitination of TBK1. Meanwhile, our results showed that the RING domain in Dtx4 was necessary for modifying TBK1 through ubiquitination.

Recent studies have demonstrated that CDK activity is crucial for virus-induced innate immune responses (*Zheng and Tang, 2022*). Reports indicate that CDKs are involved in the Toll-like receptor (TLR) signaling pathway, the nuclear factor-κB (NF-κB) signaling pathway, and the JAK-STAT signaling pathway. For instance, CDK8 and/or CDK19 enhanced the transcription of inflammatory genes, such as IL-8 and IL-10, in cells following TLR9 stimulation (*Yamamoto et al., 2017*). CDKs and NF-κB establish a remarkable paradigm where CDKs can act directly on substrate proteins rather than depending solely on transcriptional control (*Perkins et al., 1997*). It has been reported that CDK1 serves as a positive regulator of the IFN-I signaling pathway, facilitating STAT1 phosphorylation, which subsequently boosts the expression of ISGs (*Wu et al., 2016*). Furthermore, inhibiting CDK activity has been shown to obstruct STAT phosphorylation, proinflammatory gene activation, and ISG mRNA induction in response to SeV infection (*Cingöz and Goff, 2018*). It is important to note that no evidence suggests the involvement of CDKs in RLR signaling pathways. This study has shown that fish CDK2 functions as a negative regulator of the key kinase TBK1, which is involved in the RLR signaling pathway. Variations in CDK2 activity during different phases of the cell cycle may lead to changes in the expression and function of TBK1. Our findings suggest that heightened CDK2 activity may suppress TBK1 expression, thereby hindering the cell's capacity to produce IFN. Conversely, during the late phase of the cell cycle or in an inhibited state, TBK1 expression may rise, enhancing IFN synthesis and release. In summary, CDK2 is involved in intracellular signaling by modulating TBK1 levels and IFN production, affecting the cellular immune response and cycle regulation—two processes that are notably distinct at various stages of the cell cycle. A better understanding of the relationship between CDK2 and RLR signaling pathways will enhance our grasp of the regulatory mechanisms of CDKs in antiviral innate immunity. In addition, we now briefly propose a model wherein CDK2 activity during the S phase may suppress TBK1-mediated IFN production to allow viral replication, while CDK2 inhibition (e.g. in G1) may enhance IFN responses. This hypothesis will be the subject of our future work, including cell cycle synchronization experiments and time-course analyses of CDK2 activity and IFN output during infection.

CDK2 is a multifunctional kinase involved in many critical cellular processes, including cell cycle progression, differentiation, cancer, immunity, etc. (*Tadesse et al., 2020*). To date, there has been limited research conducted on fish CDK2 in the regulation of cell cycle progression. The details are as follows: It has been reported that the kinase activity of goldfish CDK2 significantly increases during oocyte maturation (*Hirai et al., 1992*). Furthermore, UHRF1 phosphorylation by cyclin A2/CDK2 is crucial for zebrafish embryogenesis (*Chu et al., 2012*). Additionally, a novel CDK2 homolog has been identified in Japanese lamprey, which plays a crucial role in apoptosis (*Xu et al., 2019*). Red grouper nervous necrosis virus (RGNNV) infection activates the p53 pathway, leading to the upregulation of p21 and downregulation of cyclin E and CDK2, which forces infected cells to remain in the G1/S replicative phase (*Mai et al., 2018*). In addition, we selected representative species from each of the six major vertebrate groups and compared their CDK2 protein sequences, discovering that they are over 90% similar to one another. This suggests that the function of CDK2 may be conserved to some extent across vertebrates. Furthermore, CDK2 inhibition has been shown to enhance anti-tumor immunity by increasing the IFN response to endogenous retroviruses (*Chen et al., 2022*). Here, we reveal the role of CDK2 in modulating the RLR signaling pathway during innate immunity. Upon infection with SVCV, CDK2 expression was induced. However, the precise upstream signaling pathways that regulate

CDK2 during viral infection remain to be fully elucidated. It is hypothesized that viral RNA sensors may activate transcription factors that bind to the cdk2 promoter; however, further investigation is required to confirm this. CDK2 deficiency or knockdown enhanced the antiviral response both in vitro and in vivo, while CDK2 overexpression promoted viral replication. Thus, our study identifies CDK2 as a negative regulator of antiviral immune responses in addition to its well-studied function in cell cycle regulation. Meanwhile, the E3 ubiquitin ligase Dtx4 is also required to regulate antiviral immune responses. Furthermore, evidence is presented demonstrating that CDK2 enhances the interaction between Dtx4 and TBK1, thus suggesting that CDK2 functions as a scaffold protein to facilitate the formation of a ternary complex. However, further study is required to ascertain the precise structural basis of this interaction, including whether CDK2's kinase activity is required.

In conclusion, our results identified a novel function of CDK2 in the negative regulation of TBK1-mediated IFN production. CDK2 interacted with TBK1 and recruited the E3 ubiquitin ligase Dtx4 to facilitate the K48-linked polyubiquitination at Lys567 residues in TBK1, eventually leading to the proteasomal degradation of TBK1. Our findings have revealed a previously unrecognized role for CDK2 in regulating immune homeostasis, providing molecular insight into the mechanisms through which CDK2–DTX4 targets TBK1 for ubiquitination and degradation.

## Materials and methods

### Fish, cells, and viruses

Mature zebrafish individuals 2.5 months after hatching (0.4±0.1 g) were selected in this study. AB line wild-type zebrafish (*Danio rerio*) and *cdk2* mutant zebrafish (CZ1442: *cdk2*^ihb488/+) were obtained from the China National Zebrafish Resource Center (CZRC) and bred using standardized procedures. In accordance with ethical requirements and national animal welfare guidelines, all experimental fish were required to undergo a two-week acclimatization period in the laboratory and have their health assessed prior to the study. Only fish that appeared healthy and were mobile were used for scientific research. Zebrafish embryonic fibroblast cells (ZF4) (RRID:CVCL_3275) (American Type Culture Collection, ATCC) were cultured in Ham's F-12 medium (Thermo Scientific, 11765054) supplemented with 10% fetal bovine serum (FBS) (Vivacell, C04001-500) at 28 °C and 5% $CO_2$. *Epithelioma papulosum cyprini* (EPC) cells (RRID:CVCL_6E02) and *Ctenopharyngodon idellus* kidney (CIK) cells (RRID:CVCL_CV32) were obtained from the Chinese Culture Collection Centre for Type Cultures (CCTCC), Gibel carp brain (GiCB) cells (RRID:CVCL_CW64) were provided by Ling-Bing Zeng (Yangtze River Fisheries Research Institute, Chinese Academy of Fishery Sciences). These cells were maintained at 28 °C in 5% $CO_2$ in medium 199 (Thermo Scientific, 11150067) supplemented with 10% FBS. THP1 cells (human acute monocytic leukemia cells) (RRID:CVCL_0006) originally obtained from ATCC were maintained in RPMI 1640 medium (Thermo Scientific, 11875085) supplemented with 10% FBS. All cell lines were routinely tested for mycoplasma. SVCV was propagated in EPC cells until a cytopathogenic effect (CPE) was observed, and then cell culture fluid containing SVCV was harvested and centrifuged at $4 \times 10^3$ g for 20 min to remove the cell debris, and the supernatant was stored at –80 °C until used. Grass carp reovirus (GCRV, strain 873, group I) and vesicular stomatitis virus (VSV) was provided by Prof. Wuhan Xiao (Institute of Hydrobiology, Chinese Academy of Sciences). GCRV was propagated in CIK cells and harvested in a similar way to SVCV. Cyprinid herpesvirus 2 (CyHV2, obtained from Yancheng city, Jiangsu province, China) was provided by Prof. Liqun Lu (Shanghai Ocean University). CyHV-2 was propagated in GICB cells and harvested in a similar way to SVCV.

### Plasmid construction and reagents

The sequence of zebrafish CDK2 (GenBank accession number: NM_213406.1) was obtained from the National Centre for Biotechnology Information (NCBI) website. CDK2 was amplified by polymerase chain reaction (PCR) using cDNA from adult zebrafish tissues as a template and cloned into the expression vector pCMV-HA (Cat# 631604) or pCMV-Myc (Cat# K6003-1) (Clontech) vectors. Zebrafish MAVS (NM_001080584.2), TBK1 (NM_001044748.2) and the truncated mutants of TBK1, Dtx4 (XM_002660524.5) and the truncated mutants of Dtx4, Trim11 (XM_021470074.1), Traip (NM_205607.1), Socs3a (NM_199950.1), and GAPDH (NM_001115114.1) were cloned into pCMV-Myc and pCMV-Tag2C vectors. The short hairpin RNA of Pimephales promelas CDK2 (XM_039663387.1) and Dtx4 (XM_039689084.1) were designed by BLOCK-iT RNAi Designer and cloned into the

pLKO.1-TRC Cloning vector. For subcellular localization experiments, CDK2 was constructed onto pEGFP-N3 (Clontech), while MAVS and TBK1 were constructed onto pCS2-mCherry (Clontech). The plasmids containing zebrafish IFNφ1pro-Luc and ISRE-Luc in the pGL3-Basic luciferase reporter vector (Promega) were constructed as described previously. The *Renilla* luciferase internal control vector (pRL-TK) was purchased from Promega. The ubiquitin mutant expression plasmids Lys-48 and Lys-63 (all lysine residues were mutated except Lys-48 or Lys-63) were ligated into the pCMV-HA vectors named Ub-K48O-HA and Ub-K63O-HA. The primers and their sequences used in this study are shown in *Supplementary file 1*, and all constructs were confirmed by DNA sequencing. Polyinosinic-polycytidylic acid (poly I:C) was purchased from Sigma-Aldrich (P0913) used at a final concentration of 1 µg/µl. MG132 (M7449), 3-Methyladenine (3-MA) (M9281), Chloroquine (CQ) (C6628) were obtained from Sigma-Aldrich. Bafilomycin A1 (Baf-A1) (S1413) was obtained from Selleck.

## Transcriptomic analysis

Total RNA was extracted using the TRIzol method and assessed for RNA purity and quantification using a NanoDrop 2000 spectrophotometer (Thermo Scientific, Waltham, U.S.A.), and RNA integrity was assessed using an Agilent 2100 Bioanalyzer (Agilent Technologies, Santa Clara, U.S.A.). Transcriptome sequencing and data analysis were performed by OE Biotech (Shanghai, China). The raw sequencing data was submitted to the NGDC (National Genomics Data Center) (GSA accession number: CRA008409).

## Transient transfection and virus infection

EPC cells were transfected in 6-well and 24-well plates using transfection reagents from FishTrans (MeiSenTe Biotechnology) according to the manufacturer's protocol. Antiviral assays were performed in 24-well plates by transfecting EPC cells with the plasmids shown in the figure. At 24 hr post-transfection, cells were infected with SVCV at a multiplicity of infection (MOI = 0.001), GCRV (MOI = 0.001), and CyHV-2 (MOI = 0.1). After 48 or 72 hr, supernatant aliquots were harvested for detection of virus titers, the cell monolayers were fixed by 4% paraformaldehyde (PFA) and stained with 1% crystal violet for visualizing CPE. For virus titration, 200 µl of culture medium were collected at 48 hr post-infection and used for detection of virus titers according to the method of Reed and Muench. The supernatants were subjected to threefold or 10-fold serial dilutions and then added (100 µl) onto a monolayer of EPC cells cultured in a 96-well plate. After 48 or 72 hr, the medium was removed and the cells were washed with PBS, fixed by 4% PFA, and stained with 1% crystal violet. The virus titer was expressed as 50% tissue culture infective dose (TCID$_{50}$/ml). For viral infection, fish were anesthetized with methanesulfonate (MS-222) and intraperitoneally (i.p.) injected with 5 µl of M199 containing SVCV ($5 \times 10^8$ TCID$_{50}$/ml). The i.p. injection of PBS was used as a mock infection. Then the fish were migrated into the aquarium containing new aquatic water.

## Luciferase activity assay

EPC cells were cultured overnight in 24-well plates and then co-transfected with the expression plasmid and luciferase reporter plasmid. The cells were infected with SVCV or transfected with poly I:C for 24 hr prior to harvest. To ensure that the same total amount of DNA was transfected in each well, pCMV-HA empty vector was used. At 24 hr post-stimulation, cells were washed with phosphate-buffered saline (PBS) and lysed for measuring luciferase activity by the Dual-Luciferase Reporter Assay System (Promega) according to the manufacturer's instructions. Firefly luciferase activity was normalized based on the Renilla luciferase activity.

## RNA extraction, reverse transcription, and quantitative PCR (qPCR)

The RNA was extracted using TRIzol reagent (Invitrogen), and first-strand cDNA was synthesized with a PrimeScript RT kit with gDNA Eraser (Takara). qPCR was performed on the CFX96 Real-Time System (Bio-Rad) using SYBR green PCR Master Mix (Yeasen). PCR conditions were as follows: 95 °C for 5 min and then 40 cycles of 95 °C for 20 s, 60 °C for 20 s, and 72 °C for 20 s. All primers used for qPCRs are shown in *Supplementary file 1*, and *β-actin* gene was used as an internal control. The relative fold changes were calculated by comparison to the corresponding controls using the $2^{-\Delta\Delta Ct}$ method (where CT was the threshold cycle).

## Co-immunoprecipitation (Co-IP) assay

EPC cells were cultured in 10 cm² dishes overnight and transfected with 10 µg of plasmid, as shown. At 24 hr post-transfection, the medium was discarded, and the cells were washed with PBS. Then the cells were lysed in 1 ml of radioimmunoprecipitation (RIPA) lysis buffer [1% NP-40, 50 mM Tris-HCl, pH 7.5, 150 mM NaCl, 1 mM EDTA, 1 mM NaF, 1 mM sodium orthovanadate ($Na_3VO_4$), 1 mM phenyl-methylsulfonyl fluoride (PMSF), 0.25% sodium deoxycholate] containing protease inhibitor cocktail (Sigma-Aldrich) at 4 °C for 1 hr on a rocker platform. The cellular debris was removed by centrifugation at 12,000×$g$ for 15 min at 4 °C. The supernatant was transferred to a fresh tube and incubated with 20 µl anti-Flag/HA/Myc affinity gel (Sigma-Aldrich, A2220/E6779/E6654) overnight at 4 °C with constant rotating incubation. These samples were further analyzed by immunoblotting (IB). Immunoprecipitated proteins were collected by centrifugation at 5000×$g$ for 1 min at 4 °C, washed three times with lysis buffer, and resuspended in 50 µl 2x SDS sample buffer. The immunoprecipitates and whole cell lysates (WCLs) were analyzed by IB with the indicated antibodies (Abs).

## In vivo ubiquitination assay

Transfected EPC cells were washed twice with 10 mL ice-cold PBS and then digested with 1 mL 0.25% trypsin-EDTA (1x) (Invitrogen) for 2–3 min until the cells were dislodged. 100 µL FBS was added to neutralize the trypsin and the cells were resuspended into 1.5 mL centrifuge tube, centrifuged at 2000×$g$ for 5 min. The supernatant was discarded and the cell precipitations were resuspended using 1 mL PBS and centrifuged at 2000×$g$ for 5 min. The collected cell precipitations were lysed using 100 µL PBS containing 1% SDS and denatured by heating for 10 min. The supernatants were diluted with lysis buffer until the concentration of SDS was decreased to 0.1%. The diluted supernatants were incubated with 20 µL anti-Myc affinity gel overnight at 4 °C with constant agitation. These samples were further analyzed by IB. Immunoprecipitated proteins were collected by centrifugation at 5000×$g$ for 1 min at 4 °C, washed three times with lysis buffer and resuspended in 100 µL 1x SDS sample buffer.

## Immunoblot analysis

Immunoprecipitates or WCLs were analyzed as described previously. Antibodies were diluted as follows: anti-β-actin (ABclonal, AC026) at 1:3000, anti-Flag (Sigma-Aldrich, F1804) at 1:3000, anti-HA (Covance, MMS-101R) at 1:3000, anti-Myc (Santa Cruz Biotechnology, sc-40) at 1:3000, anti-Dtx4 (Thermo Scientific, PA5-46146) at 1:1000, anti-CDK2 (GeneTex, GTX101226) at 1:1000, and HRP-conjugated anti-mouse/rabbit IgG (Thermo Scientific, 31430/31460) at 1:5000, anti-N/P/G/TBK1/CDK2 (prepared and purified in our lab) at 1:2000.

## Immunofluorescence (IF)

EPC cells were plated onto glass coverslips in 6-well plates and infected with SVCV (MOI = 1) for 24 hr. Then the cells were washed with PBS and fixed in 4% PFA at room temperature for 1 hr and permeabilized with 0.2% Triton X-100 in ice-cold PBS for 15 min. The samples were blocked for 1 hr at room temperature in PBS containing 2% bovine serum albumin (BSA, Sigma-Aldrich). After additional PBS washing, the samples were incubated with anti-N Ab in PBS containing 2% BSA for 2–4 hr at room temperature. After being washed three times by PBS, the samples were incubated with secondary Ab Alexa Fluor 488 AffiniPure Donkey anti-Rabbit IgG (H+L) (34206ES60, 1:10,000) in PBS containing 2% BSA for 1 hr at room temperature. After additional PBS washing, the cells were finally stained with 1 µg/ml 4′, 6-diamidino-2-phenylindole (DAPI; Beyotime Institute of Biotechnology, Shanghai, China) for 10 min in the dark at room temperature. Finally, the coverslips were washed and observed with a confocal microscope under a 10x immersion objective (SP8; Leica).

## Fluorescent microscopy

EPC cells were plated onto coverslips in 6-well plates and transfected with the plasmids indicated for 24 hr. Then the cells were washed twice with PBS and fixed with 4% PFA for 1 hr. After being washed three times with PBS, the cells were stained with 1 µg/ml DAPI for 15 min in the dark at room temperature. Finally, the coverslips were washed and observed with a confocal microscope under a 63x oil immersion objective (SP8; Leica).

## Histopathology

Liver, spleen, and kidney tissues from three individuals of control or virus-infected fish at 2 days post-infection (dpi) were dissected and fixed in 10% phosphate-buffered formalin overnight. Then the samples were dehydrated in ascending grades of alcohol and embedded into paraffin. Sections at 5 μm thickness were taken and stained with hematoxylin and eosin (H&E). Histological changes were examined by optical microscopy at 40x magnification and were analyzed by the Aperio ImageScope software.

## Statistical analysis

For fish survival analysis, Kaplan-Meier survival curves were generated and analyzed by the Log-rank test. For the bar graph, one representative experiment of at least three independent experiments is shown, and each was done in triplicate. For the dot plot graph, each dot point represents one independent biological replicate. Unpaired Student's t-test was used for statistical analysis. Data are expressed as mean ± standard error of the mean (SEM). A $p$-value $<0.05$ was considered statistically significant.

## Acknowledgements

We thank Fang Zhou (Institute of Hydrobiology, Chinese Academy of Sciences) for assistance with confocal microscopy analysis and Dr. Feng Xiong (China Zebrafish Resource Center, Institute of Hydrobiology, Chinese Academy of Sciences) for assistance with qPCR analysis. This work was supported by the Strategic Priority Research Program of the Chinese Academy of Sciences (XDB0730300), National Excellent Youth Science Fund (32322086), National Natural Science Foundation of China (32073009), and the Youth Innovation Promotion Association provided funding to Shun Li. National Key Research and Development Program of China (2023YFD2400201) provided funding to Dan-Dan Chen. National Natural Science Foundation of China (32173023) provided funding to Long-Feng Lu.

## Additional information

### Funding

| Funder | Grant reference number | Author |
| --- | --- | --- |
| Strategic Priority Research Program of the Chinese Academy of Sciences | XDB0730300 | Shun Li |
| National Excellent Youth Science Fund | 32322086 | Shun Li |
| National Key Research and Development Program of China | 2023YFD2400201 | Dan-Dan Chen |
| National Natural Science Foundation of China | 32173023 | Long-Feng Lu |
| National Natural Science Foundation of China | 32073009 | Shun Li |
| Youth Innovation Promotion Association | | Shun Li |

The funders had no role in study design, data collection and interpretation, or the decision to submit the work for publication.

### Author contributions

Long-Feng Lu, Conceptualization, Data curation, Funding acquisition, Writing – original draft; Can Zhang, Bao-Jie Cui, Validation; Zhuo-Cong Li, Ke-Jia Han, Xiao-Yu Zhou, Xiao-Li Yang, Methodology; Yang-Yang Wang, Xiao Xu, Na Xu, Investigation; Chu-Jing Zhou, Yue Wu, Software; Dan-Dan Chen,

Software, Funding acquisition; Xiyin Li, Li Zhou, Supervision; Shun Li, Conceptualization, Supervision, Funding acquisition, Writing – review and editing

### Author ORCIDs
Shun Li ⬤ https://orcid.org/0000-0002-3629-9900

### Ethics
The experiments involved in this study were conducted in compliance with ethical regulations. The fish experiments were carried out under the guidance of the European Union Guidelines for the Handling of Laboratory Animals (2010/63/EU) and approved by the Ethics Committee for Animal Experiments of the Institute of Aquatic Biology of the Chinese Academy of Sciences (No. 2023-068).

Reviewer #1 (Public review): https://doi.org/10.7554/eLife.98357.4.sa1
Reviewer #2 (Public review): https://doi.org/10.7554/eLife.98357.4.sa2
Author response https://doi.org/10.7554/eLife.98357.4.sa3

---

## Additional files

### Supplementary files
Supplementary file 1. Primers used in this study.

MDAR checklist

### Data availability
RNA sequence data were deposited in NGDC (National Genomics Data Center) (GSA accession number: CRA008409). All data generated or analyzed during this study are included in the manuscript and supporting files; source data files have been provided for all figures.

The following dataset was generated:

| Author(s) | Year | Dataset title | Dataset URL | Database and Identifier |
|---|---|---|---|---|
| Li Z | 2024 | Aquatic virus | https://ngdc.cncb.ac.cn/gsa/search?searchTerm=CRA008409 | National Genomics Data Center, CRA008409 |

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
