## [Editor Report · eLife Assessment]

This **valuable** study uses zebrafish as a model to reveal a role for the cell cycle protein kinase CDK2 as a negative regulator of type I interferon signaling. The evidence supporting the authors' claims is **convincing**, including both in vivo and in vitro investigative approaches that corroborate a role for CDK2 in regulating TBK1 degradation. In this latest version, the authors included data addressing a concern raised by the reviewer in the previous peer review round. This work will interest cell biologists, immunologists, and virologists.

---

## [Referee Report · Reviewer #1 (Public review)]

Summary:

The authors set out to evaluate the regulation of interferon (IFN) gene expression in fish, using mainly zebrafish as a model system. Similar to more widely characterized mammalian systems, fish IFN is induced during viral infection through the action of the transcription factor IRF3 which is activated by phosphorylation by the kinase TBK1. It has been previously shown in many systems that TBK1 is subjected to both positive and negative regulation to control IFN production. In this work, the authors find that the cell cycle kinase CDK2 functions as a TBK1 inhibitor by decreasing its abundance through recruitment of the ubiquitinylation ligase, Dtx4, which has been similarly implicated in the regulation of mammalian TBK1. Experimental data are presented showing that CDK2 interacts with both TBK1 and Dtx4, leading to TBK1 K48 ubiqutinylation on K567 and its subsequent degradation by the proteasome.

Strengths:

The strengths of this manuscript are its novel demonstration of the involvement of CDK2 in a process in fish that is controlled by different factors in other vertebrates and its clear and supportive experimental data.

Weaknesses:

The weaknesses of the study include the following. (1) It remains unclear how CDK is regulated during viral infection and how it specifically recruits E3 ligase to TBK1. The authors find that its abundance increases during viral infection, an unusual finding given that CDK2 levels are often found to be stable. How this change in abundance might affect cell cycle control was not explored. (2) The implications and mechanisms for a relationship between the cell cycle and IFN production will be a fascinating topic for future studies. In particular, it will be critical to determine if CDK2 catalytic activity is required. An experiment with an inhibitor suggests that this novel action of CDK2 is kinase independent, but the lack of controls showing the efficacy of the inhibitor prevents a firm conclusion. It will also be critical to determine if there is a role for cyclins in this process or if there is competition for binding between TBK1 and cyclin and, if so, if this has an impact on the cell cycle. Likewise, an impact of CDK2 induction by virus infection on normal cell cycling will be important to investigate.

---

## [Referee Report · Reviewer #2 (Public review)]

Summary:

In this paper, the authors describe a novel function involving the cell cycle protein kinase CDK2, which binds to TBK1 (an essential component of the innate immune response) leading to its degradation in a ubiquitin/proteasome-dependent manner. Moreover, the E3 ubiquitin ligase, Dtx4, is implicated in the process by which CDK2 increases the K48-linked ubiquitination of TBK1. This paper presents intriguing findings on the function of CDK2 in lower vertebrates, particularly its regulation of IFN expression and antiviral immunity.

Strengths:

(1) The research employs a variety of experimental approaches to address a single question. The data are largely convincing and appear to be well executed.

(2) The evidence is strong and includes a combination of in vivo and in vitro experiments, including knockout models, protein interaction studies, and ubiquitination analyses.

(3) This study significantly impacts the field of immunology and virology, particularly concerning the antiviral mechanisms in lower vertebrates. The findings provide new insights into the regulation of IFN expression and the broader role of CDK2 in immune responses. The methods and data presented in this paper are highly valuable for the scientific community, offering new avenues for research into antiviral strategies and the development of therapeutic interventions targeting CDK2 and its associated pathways.

---

## [Author Response]

The following is the authors’ response to the previous reviews.

**Reviewer #1 (Public Review):**
The weaknesses of the study include the following.(1) It remains unclear how CDK is regulated during viral infection and how it specifically recruits E3 ligase to TBK1.

We would like to express our gratitude to the reviewer for highlighting this significant issue. The present study demonstrates that CDK2 expression is significantly upregulated upon SVCV infection in multiple fish tissues and cell lines (see Fig. 1C-F), thus suggesting that viral infection triggers CDK2 induction. However, the precise upstream signaling pathways that regulate CDK2 during viral infection remain to be fully elucidated. It is hypothesized that viral RNA sensors may activate transcription factors that bind to the cdk2 promoter; however, further investigation is required to confirm this. We have added a sentence in the Discussion (Lines 409-412) acknowledging this as a limitation and a focus for future work, suggesting potential involvement of viral sensor pathways.

With regard to the mechanism by which CDK2 recruits the E3 ligase Dtx4 to TBK1, evidence is provided that CDK2 directly interacts with both TBK1 (via its kinase domain) and Dtx4 (see Fig. 4F-I, 6A-C). Furthermore, evidence is presented demonstrating that CDK2 enhances the interaction between Dtx4 and TBK1 (Fig. 6D), thus suggesting that CDK2 functions as a scaffold protein to facilitate the formation of a ternary complex. However, further study is required to ascertain the precise structural basis of this interaction, including whether CDK2's kinase activity is required. We have added a note in the Discussion (Lines 417-421) acknowledging this limitation and proposing future structural studies to elucidate the precise binding interfaces.

(2) The implications and mechanisms for a relationship between the cell cycle and IFN production will be a fascinating topic for future studies.

We concur with the reviewer's assertion that the interplay between cell cycle progression and innate immunity constitutes a promising and under-explored research domain. Whilst the present study concentrates on the function of CDK2 in antiviral signaling, independent of its cell cycle functions, it is acknowledged that CDK2's activity is cell cycle-dependent. It is hypothesized that CDK2 may function as a molecular link between cell proliferation and immune responses, particularly in light of the observation that viral infections frequently modify host cell cycle progression. In the Discussion (lines 387-391), we now briefly propose a model wherein CDK2 activity during the S phase may suppress TBK1-mediated IFN production to allow viral replication, while CDK2 inhibition (e.g., in G1) may enhance IFN responses. This hypothesis will be the subject of our future work, including cell cycle synchronization experiments and time-course analyses of CDK2 activity and IFN output during infection.

**Reviewer #1 (Recommendations for the authors):**
(1) A control showing that the CDK2 inhibitor blocked kinase activity would be appropriate.

We thank the reviewer for this suggestion. We have performed experiments using the CDK2-specific inhibitor SNS-032. As shown in the Author response image 1, the treatment of EPC cells with SNS-032 (2 µM) still affect TBK1 expression. However, the selection of this inhibitor was based on literature references (ref. 1 and 2), and it is uncertain whether it directly inhibits the kinase activity of CDK2. However, our result demonstrated that CDK2 retains the capacity to degrade TBK1 even in the absence of its kinase domain (Fig. 6I), yielding outcomes that are consistent with this inhibitor.

References:

(1) Mechanism of action of SNS-032, a novel cyclin-dependent kinase inhibitor, in chronic lymphocytic leukemia. Blood. 2009 May 7;113(19):4637-45.

(2) SNS-032 is a potent and selective CDK 2, 7 and 9 inhibitor that drives target modulation in patient samples. Cancer Chemother Pharmacol. 2009 Sep;64(4):723-32.